# CBN-VAE: A Data Compression Model with Efficient Convolutional Structure for Wireless Sensor Networks

**DOI:** 10.3390/s19163445

**Published:** 2019-08-07

**Authors:** Jianlin Liu, Fenxiong Chen, Jun Yan, Dianhong Wang

**Affiliations:** School of Mechanical Engineering and Electronic Information, China University of Geosciences, Wuhan 430074, China

**Keywords:** wireless sensor networks, data compression, variational autoencoder, downsampling-convolutional restricted boltzmann machine

## Abstract

Data compression is a useful method to reduce the communication energy consumption in wireless sensor networks (WSNs). Most existing neural network compression methods focus on improving the compression and reconstruction accuracy (i.e., increasing parameters and layers), ignoring the computation consumption of the network and its application ability in WSNs. In contrast, we pay attention to the computation consumption and application of neural networks, and propose an extremely simple and efficient neural network data compression model. The model combines the feature extraction advantages of Convolutional Neural Network (CNN) with the data generation ability of Variational Autoencoder (VAE) and Restricted Boltzmann Machine (RBM), we call it CBN-VAE. In particular, we propose a new efficient convolutional structure: Downsampling-Convolutional RBM (D-CRBM), and use it to replace the standard convolution to reduce parameters and computational consumption. Specifically, we use the VAE model composed of multiple D-CRBM layers to learn the hidden mathematical features of the sensing data, and use this feature to compress and reconstruct the sensing data. We test the performance of the model by using various real-world WSN datasets. Under the same network size, compared with the CNN, the parameters of CBN-VAE model are reduced by 73.88% and the floating-point operations (FLOPs) are reduced by 96.43% with negligible accuracy loss. Compared with the traditional neural networks, the proposed model is more suitable for application on nodes in WSNs. For the Intel Lab temperature data, the average Signal-to-Noise Ratio (SNR) value of the model can reach 32.51 dB, the average reconstruction error value is 0.0678 °C. The node communication energy consumption can be reduced by 95.83%. Compared with the traditional compression methods, the proposed model has better compression and reconstruction accuracy. At the same time, the experimental results show that the model has good fault detection performance and anti-noise ability. When reconstructing data, the model can effectively avoid fault and noise data.

## 1. Introduction

As one of the core technologies of the Internet of Things (IOT), wireless sensor networks (WSNs) have the characteristics of limited capacity and dense node distribution, and they have become a focus of academic and industrial circles. WSNs usually deploy a large number of micro-sensor nodes in the monitoring area, and form a multi-hop self-organizing communication network through wireless communication technology to realize cooperative sensing of multi-sensor nodes. By collecting and processing various information of objects monitored in the network coverage area, users can obtain various types of monitoring data in the WSNs deployment area at any time. WSNs have been widely used in military reconnaissance, medical monitoring, smart home, intelligent transportation and industrial and agricultural automation [1]. Often, sensor nodes are powered by battery and deployed in an unattended hostile environment with high density. In most cases, the radio transceiver onboard sensor nodes is the main cause of energy consumption. The energy problem has always been a bottleneck which limits the widespread use of WSNs. Therefore, reducing communication energy consumption has become one of the research hotspots of WSNs.

Recently, many works were accomplished to propose data transmission energy reduction scheme for WSNs. In order to increase the life of the sensor node, it is necessary to reduce the energy that the node uses to transmit data. To achieve this goal, data aggregation and data compression are the most common methods. Data aggregation focuses on observing the redundancy between data generated from neighboring sensors, and it eliminates redundant data transmission by processing of data accumulated from multiple sensors [2,3,4]. The basic idea of data aggregation is to aggregate the samples of multi-sensors with a certain degree of redundancy rather than transmit original sensing data. In data aggregation, every sensor node sends the data packets of the same size and number at various instants. Every data packet holds the value denoting the in-network estimation known as the partial rate for the subsequent midway node in the aggregation tree. Although data aggregation reduces the amount of data transferred, the calculation of redundant samples can result in the loss of important data. In addition, data aggregation requires a high-quality data transmission network, and the anti-noise ability is weak. At the same time, the aggregation results are greatly affected by the spatio-temporal correlation. Encoding data using an encoding method is another solution. The sensor LZW (S-LZW) is an algorithm that performs compression based on a dictionary [5]. The static Huffman algorithm utilizes a predefined probability table to take advantage of Huffman compression [6]. The adaptive Huffman algorithm uses a tree model representation in which nodes and leaves represent transmitted symbols [7]. At the beginning of the transfer, the statistics of the source sequence are unknown. As the transfer progresses, the new node will be added to the tree and the tree will be reconfigured. Reference [8] proposes an adaptive dictionary-based dynamic algorithm. The algorithm utilizes temporal data correlation and Huffman compression. Reference [9] completes the work of the Z-order based data compression scheme (Z compression). However, coding usually focuses on repetitive data, making compression close to lossless compression, often not reaching extremely high compression ratios.

Data compression can effectively reduce the number of data and the communication energy consumption for WSNs. Some data compression algorithms focus on data reduction based on time series approximation [10,11,12]. These works convert data samples into a set of coefficients to simplify data representation, such as FFT and WT [13,14]. The performance of this algorithms depends on the number of coefficients needed to encode input data. The more the number of algorithm coefficients, the better the performance, but the calculation energy consumption will become higher. The Lightweight Temporal Compression (LTC) algorithm proposed in Reference [15] is an efficient and simple lossy compression technique for the context of habitat monitoring. LTC introduces a small amount of error into each reading bounded by a control knob: the larger the bound on this error, the greater the saving by compression. In Reference [16], Marcelloni et al. proposed a purposely adapted version of the Differential Pulse Code Modulation (DPCM) scheme to compress the sensing data (in this paper, we call it DPCM-o), and it has better compression than LTC. In Reference [17], Donoho et al. proposed the compressed sensing (CS) method which provides a new direction for data compression in WSNs. The CS method can use the fewer measurements to recover large volumes of original data when the original data are sparse in a basis. CS can greatly reduce the cost of the system due to the use of sparse binary matrices [18]. CS requires that the signal be sparse or compressible at a certain level, otherwise the signal cannot be reconstructed. There are many signal recovery algorithms for fast reconstruction and reliable accuracy, such as the basis pursuit, the orthogonal matching pursuit (OMP), and the stagewise OMP (StOMP) [19]. BP has high computation complexity, and cannot be used for large-scale applications. OMP and StOMP adopt a bottom-up approach in signal recovery, and their complexity levels are much lower than that of BP. However, they require more measurements and have a lack of recovery guarantee.

Machine learning (ML) is a technique for artificial intelligence, which has excellent mathematical fitting ability. In recent years, convolutional neural networks (CNNs) have demonstrated amazing abilities in various fields which has promoted the wide application of ML in various fields. Variational Autoencoder (VAE) and Restricted Boltzmann Machine (RBM) are data generation models designed by ML [20,21]. They use computational methods to improve model performance by detecting and describing consistency and patterns in training data [22]. CNNs can extract deeper and richer data hiding information through multiple layers of iterative convolution. Principal Component Analysis (PCA) algorithm is a dimensionality reduction technique from machine learning, which can be used to compress cluster data of cluster heads of WSNs [23]. In Reference [24], the authors described the use of PCA in WSNs employing supervised and unsupervised compression models. Reference [25] combined PCA compression with data aggregation to combine information from many sensor nodes. In Reference [26], the authors discuss the combination of ML methods and CS, which uses feed-forward deep neural network structures to aid CS signal reconstruction. In Reference [27], the authors propose a data compression algorithm that combines stacked autoencoder (SAE) with clustered routing protocol. In Reference [28], the authors use an unsupervised neural network scheme to analyze the data temporal or spatial correlation. Reference [29] developed an optimization technique using the back-propagation neural network to search for an optimal compression ratio to improve the quality of the signal and to improve the energy efficiency of the router node. Reference [30] studied the combination of RBM and autoencoder and proposed the Stacked RBM-AE compression scheme. The large deep convolutional networks have also been applied to data compression, but most of the research is currently limited to the field of image compression. For WSNs, most people use RBM or fully connected layers which only require a small amount of computation to compress sensing data, and few people study using convolutional networks to compress sensing data. An important reason for the lack of research is that the computational consumption of deep convolutional networks is large and it is difficult to apply to sensor nodes with limited computing power. In Reference [31], the authors used deep convolutional networks to compress electrocardiogram signals (ECG), but the network requires a lot of computation.

In this paper, we explore how to apply CNNs to sensor data compression in WSN. The main problem we solve is how to reduce the computational consumption of the CNNs while ensuring that the performance of the network does not degrade. Reference [32] develops the Convolutional RBM (CRBM) structure to achieve object detection and obtained good detection results. CRBM respects the spatial structure of the image by sharing weights. We get inspiration from the design of CRBM and design an efficient convolution structure Downsampling-Convolutional RBM (D-CRBM). D-CRBM reduces network parameters by weight sharing and undirected graph characteristics of RBM, and uses downsampling to reduce the computational consumption of network. We combine D-CRBM with VAE and propose the CBN-VAE model. The model learns the data features through convolution calculation and reconstructs the data through the data generation ability of VAE and RBM. Our experiments demonstrate that the D-CRBM structure can significantly reduce the parameters and the computational consumption of the neural network without causing loss of network performance. Compared to traditional algorithms and neural network models, the proposed model has the best compression performance. We also test the transfer learning ability, fault detection ability and anti-noise ability of the proposed model. The energy analysis results prove that the computational efficiency of the model, and the model can be directly applied to the sensor nodes. We also propose a neuron pruning method to reduce redundancy in the network, which can further reduce the parameters and computational consumption of the model. The contributions of this paper are summarized as follow:We developed a new neural network data compression model named CBN-VAE for WSNs. Compared with traditional neural networks, the proposed model has fewer parameters and computation consumption with negligible accuracy loss. We proved that the proposed model is more suitable for application on nodes in WSNs by experiment.We proposed an efficient convolution structure named D-CRBM to reduce the amount of convolution operations, which can significantly reduce network parameters and computation consumption while preserving the performance of CNNs.We proposed a new method of data compression, which has better reconstruction accuracy than the traditional algorithm under the same compression ratio (CR).We proposed a new idea to judge the importance of neural network neurons. We used this idea to guide neural network pruning to further reduce network parameters and computational consumption.

The remainder of this paper is organized as follows: Section 2 describes the CBN-VAE model architecture, the structure of D-CRBM and the training algorithm for the model. Section 3 shows the experimental results. At the same time, we give suggestions for the deployment of the CBN-VAE model. Specially, we introduce our neuron importance judgment ideas and the neuron pruning algorithms. In Section 4, we summarize our model. At the same time, we give the problems that we found in experiments and the further work we are going to do.

The code of the CBN-VAE model is available online (https://github.com/LJianlin/CBN-VAE).

## 2. CBN-VAE Model Architecture

In this section, we first describe the core calculation layers that CBN-VAE model is built on which are Downsampling-Convolutional RBM. We then describe the CBN-VAE model structure and the training algorithm for the model. At the same time, we summarize the details of the model to reduce parameters and computation consumption.

### 2.1. Downsampling-Convolutional RBM

For the CBN-VAE model, Downsampling-Convolutional RBM (D-CRBM) is an important unit for learning the hidden mathematical features of original sensing data and reducing network parameters. The D-CRBM is Convolutional RBM (CRBM) with downsampling. The structure of CRBM is shown in Figure 1. The basic structure of CRBM is the standard RBM, which is an undirected graph structure consisting of two layers: input layer v and hidden layer h. The difference between CRBM and standard RBM is the method of calculating the neuron states of the input layer and the hidden layer. For standard RBM, the neuron states of the input layer and the hidden layer are obtained by directly multiplying the input and weight matrix. Since the number of weight matrices is 1, the neuron state is unique. In the CRBM, the method of calculating the neuron state of the input layer and the hidden layer is the convolution calculation. The standard RBM uses only one weight matrix to calculate the neuron state, and the CRBM uses multiple convolution kernels to calculate. However, after multiple convolution kernels are calculated, multiple corresponding neuron states are generated, which is contrary to one state result of the standard RBM. Therefore, we need to process the convolution result. We concatenate multiple convolution results into one and use it as the neuron state. This preserves the undirected graph structure of the CRBM while minimizing the loss of convolutional useful information because we retain the results for all convolution kernels. At the same time, the undirected graph feature of CRBM allows the CRBM to perform bidirectional transmission of information, that is, the convolution input can also be calculated from the convolution results, which is not possible with the standard convolutional layer. Compared to the standard convolutional layer from the input and output dimension, for a standard convolution layer with an input size of $X_{w}\times X_{h}\times C_{i}$ and the convolution kernel size of $\;K_{w}\times K_{h}\times C_{i}\times C_{o}$, the size of the convolution result is $\;O_{w}\times O_{h}\times C_{o}\;$ where $\;O_{w}=X_{w}-K_{w},\;O_{h}=X_{h}-K_{h}$ and the convolution strides are 1. For the CRBM with the same parameters, the size of the CRBM output is $\;O_{w}^{c}\times O_{h}\times 1$ where $\;O_{w}^{c}=O_{w}\times C_{o}$. The CRBM and the standard convolution have the same output values, but the the output dimensions are different.

The downsampling of D-CRBM is implemented by the convolution stride and max-pooling. The structure of D-CRBM is shown in Figure 2. In the D-CRBM, we set the convolution stride to $K_{w}$ where $K_{w}\;$ is the width of the convolution kernel. This operation can reduce the redundancy caused by the convolution kernel repeated calculation of the same data. Then we downsample the convolution result by using max-pooling, the pooling size is  2×1. Take 1-dimensional sensing data (time sequences) as an example. For the D-CRBM with an input size of $X_{w}\times 1\times 1$ and the convolution kernel size of $\;K_{w}\times 1\times 1\times C_{o}$, the size of the convolution result is $\;O_{w}^{d}\times 1\times 1\;$ where $\;O_{w}^{d}=C_{o}\times \frac{X_{w}-K_{w}}{2*K_{w}}$. Compared with the standard convolution which have the same parameters, the D-CRBM gets a reduction in computation of $2*K_{w}\;$ in the layer-wise. After max-pooling, we construct an index matrix to store the position index of the max-pooling results. The index matrix is used to recover network data when reconstructing data. The index matrix is a binarization matrix which consists of 0, 1. The usage of the index matrix is shown in Figure 3. The index matrix the same size as the max-pooling output. For the reconstruction of max-pooling, the original data is sequentially restored according to the value in the max-pooling output and the corresponding index value in the index matrix. The D-CRBM and the standard RBM are both undirected graph models, this means that the convolution kernels used to generate the hidden layer from the input can also be used to calculate the original input from the hidden layer. Therefore, for the reconstruction of the D-CRBM (shown in Figure 2), we use the same convolution kernels to get the original convolution input (i.e., deconvolution operation).

### 2.2. Model Structure and Training

The CBN-VAE model is a hybrid model which the calculation unit is D-CRBM and the model framework is VAE. Specifically, the CBN-VAE model consists of two parts: an encoder and a decoder. The encoder learns the hidden mathematical features of the original sensing data and compresses the sensing data, and the decoder reconstructs the sensing data based on the hidden mathematical features. The structure of the CBN-VAE model is shown in Figure 4, and the details of layers and parameters used for the CBN-VAE model is shown in Table 1. For a time sequence input of sensing data with size of 120 × 1, after encoding by the encoder, the data is compressed to a size of 5 × 1. The decoder reconstructs the compressed data back to the size of 120 × 1. The activation functions used by all layers are Rectified Linear Unit (ReLU) nonlinearity.

In the encoder, the model first extracts the hidden mathematical features of the input data through a combination of multiple layers of D-CRBM and max-pooling and encodes the input data by this way. We designed a fully connected layer for the output of the feature after the features were extracted. Then, we used the variational sampling method of VAE to design two fully connected layers to calculate the mean vector and the standard deviation vector of the output of the last fully connected layer. We sampled from the standard deviation vector and then added it to the mean vector to get the final output of the encoder. For a time sequence input of sensing data with size of 120 × 1, after extracting features by the D-CRBM and max-pooling, the features were represented as a vector of size of 54 × 1 × 1. Then we used the fully connected layer to convert this vector into a 2-dimensional output of size 27 × 1 and use the 2-dimensional output to calculate the mean vector and standard deviation vector of the variational sampling method. The mean vector and the standard deviation vector were added to get our final compressed data with size of 5 × 1.

In the decoder, by using the undirected graph characteristics of the D-CRBM, we built the decoding network using the parameters of the corresponding network in the encoder. Unlike the encoder network, we built a new fully connected layer in the decoder to recover the data after the variational sampling. The parameters of the other layers in the decoder network were the same as the corresponding layers in the encoder network. Combining the encoder and the decoder, we can get the model output of the same dimension for each model input. Our goal is to make the model output as close as possible to the model input itself. That is, after the model input is encoded, it is possible to recover as much of the original information as possible by decoding.

The methods we used to reduce parameters and computation consumption of CBN-VAE model are summarized as follow:Change the convolution output feature map dimension. This method can significantly reduce the number of convolution kernel parameters for the next convolutional layer. For a convolutional layer with output size of $\;O_{w}\times O_{h}\times C_{o}$, the parameters of next convolutional layer convolution kernel should be $K_{s}\times C_{o}\times K_{n}\;$ where $K_{s}$ is the size of kernel and $K_{s}$ is the number of kernel. After we turn the output to $\;O_{w}^{c}\times O_{h}\times 1$ where $\;O_{w}^{c}=O_{w}\times C_{o}$, the parameters of next convolutional layer convolution kernel should be $K_{s}\times 1\times K_{n}$. Compared with standard convolution, this method can reduce the parameters and computation consumption of $C_{o}$ in the layer-width.Share network parameters. Parameter sharing can be divided into two aspects: one is the parameter sharing of the convolution kernel in the layer-width; and the other is the undirected graph characteristic of D-CRBM, the parameters between the encoder and decoder networks can be shared.D-CRBM. Downsampling convolution by using the variable convolution stride and max-pooling. Change the convolution stride can reduce the redundancy caused by the convolution kernel repeated calculation of the same data. The max-pooling operation can further reduce the network redundancy.

We built the CBN-VAE model by using Tensorflow, and trained the model with back-propagating (BP) algorithm [33]. Algorithm 1 shows the procedure of the training of the CBN-VAE model. The goal of the training algorithm is to reduce the loss value of the network by iteratively updating the network parameters. It first calculates the reconstructed output of the network through forward propagation and then calculates the network loss based on the loss function. The loss value is used for the calculation of the gradient in back propagation. Finally, it uses this gradient and learning rate to update the parameters.

**Algorithm 1:** Training of the Stacked CBN-VAE model.1:**Input:** A mini-batch of training data set **S**, the number of train iteration *iter*, the learning rate *α*2:  **While**
*i* < *iter*
**do**3:  Compute the reconstructed output through forward propagation4:  Compute the model loss according to (3)5:  Compute the gradient of parameters according to (4)6:  Update parameters according to (5)7:  *i* + +8:**end**

For the CBN-VAE model, the goal is to make the model output as close as possible to the model input itself. For the input data and output data of the model, we use the mean-square error to measure the difference:(1)$$mse=\frac{\sum_{i=1}^{n}{\left(x_{i}-x_{i}^{’}\right)}^{2}}{n}$$
where $x_{i}$ is the input data, $x_{i}^{’}$ is the model output.

We hope that the model will not only correctly fit the original data, but also learn the correct distribution of the original data. We assume that the original data satisfies the normal distribution. In the variational part of the model, the model calculates a mean vector and a standard deviation vector, that is, the data distribution learned by the model. In the reconstruction section, the model uses this distribution for data reconstruction. In order to make the data learned by the model reasonable, we use Kullback-Leibler (KL) divergence to constrain mean vector and standard deviation vector:(2)$$KL=-0.5\times \left(1+\log\sigma^{2}-\mu^{2}-\exp\left(\log\sigma^{2}\right)\right)$$
where $\log\sigma^{2}$ is the standard deviation vector, $\mu^{2}$ is the mean vector.

The final loss of the CBN-VAE model is defined as:(3)loss=mse+λ∗KL  
where *λ* is the equilibrium factor with a value between 0 and 1. At the same time, the addition of KL divergence is equivalent to adding a regularization term to the original mse loss, which can avoid over-fitting of the model.

During the training process, the parameters of the model are updated by their corresponding gradients and learning rates in each iteration. For the parameter $W_{i,k}$ of the *i*-th layer, the gradient is calculated as follows:(4)$$Grad_{W_{i,k}}=\frac{\partial loss}{\partial W_{i,k}}\;\;$$

In the back propagation process, the parameter $\;W_{i,k}$ update formula is:(5)$$W_{i,k}=W_{i,k}-\alpha *Grad_{W_{i,k}}\;\;$$
where α is the learning rate.

## 3. Experiments

We used the following performance criteria to evaluate the performance of the proposed model: (1) compression ratio (CR); (2) percentage RMS difference (PRD); (3) signal-to-noise ratio (SNR). The definitions and formulas of these performance criteria are as follows:

(1) Compression Ratio (CR): It is defined as the compressed data length over the size of uncompressed data.
(6)$$CR=\frac{D_{cp}}{D_{or}}\;\;$$
where $D_{or}$ is the number of bytes of all original data. $D_{cp}$ is the number of bytes of all compressed data.

(2) Percentage RMS Difference (PRD): It represents the quality of reconstructed data in the compression. The PRD value is expected to be as low as possible for a quality compression approach.
(7)PRD(%)=100×(∑i=0D−1(So(i)−Sr(i))2∑i=0D−1(So(i))2)12
where $S_{o}$ represents the original input data, $S_{r}$ represents the reconstructed data.

(3) Signal-to-Noise Ratio (SNR): SNR is the ratio of signal to noise. The SNR is calculated by:(8)$$SNR\left(dB\right)=10\log_{10}\left(\frac{\sum_{i=0}^{D-1}{\left(S_{o}\left(i\right)\right)}^{2}}{\sum_{i=0}^{D-1}{\left(S_{r}\left(i\right)-S_{o}\left(i\right)\right)}^{2}}\right)$$

We built the Tensorflow framework on the computer with Ubuntu 16.04 LTS system and NVIDIA GeForce GTX 1080 Ti as our experimental environment, and then built the CBN-VAE model by using Tensorflow. All our experiment results were obtained using this environment.

### 3.1. Dataset Preprocess

First, we used the Intel Berkeley Research Laboratory (IBRL) [34] as the experimental dataset. The IBRL dataset was collected from 54 sensor nodes at the Intel Berkeley Research Lab from 28 February, 2004 to 6 April, with a sampling interval of 31 s. The sensing data in IBRL dataset contains four categories: temperature, humidity, light and voltage. The amount of temperature data is large and the temperature change trend is intuitive, so we select the temperature data as our experimental data. Due to node failure, there were several temperature values above 100 °C and below −30 °C. Because the sensor nodes were located indoors, we first removed the apparent anomaly data in the temperature data by taking the threshold of 45 °C and −5 °C by a priori knowledge. Then we removed most of the noise in the data by retaining those data the values of which were in $\left[x_{mean}-3*x_{std},\;x_{mean}-3*x_{std}\right]$. $x_{mean}$ denotes the mean value of the sample data, $x_{std}$ denotes the standard deviation of the sample data.

In order to reduce the difference between the input sensing data and converge the algorithm more quickly, we mapped the original sensing data to range [0, 1] by using max-min normalization. For this processed sensing data, we divided each node data into two parts: training set and test set, and the split ratio was 8 to 2.

Then, we tested the performance of the proposed model on different real-world WSN datasets (Argo [35], ZebraNet [36] and CRAWDAD [36]) to evaluate the generalization and robustness of the proposed model.

### 3.2. Compression and Reconstruction

For the CBN-VAE model, the CR is 24 (as shown in Figure 4, the input dimension is 120 and the output dimension is 5 with an identical data type of input and output). In the following experiments, the performance results of the model are tested with CR of 24.

The learning rate is a hyperparameter that is usually set by the experimenter and can directly affect the final performance and convergence speed of the model. However, the difficulty of adjusting parameters is a big problem in neural networks. For example, for stochastic gradient descent (SGD) algorithm [37], at the beginning we hope that the parameter adjustment is bigger, the learning rate is larger, the convergence is accelerated. In the later stages of training the learning rate, we hope that the learning rate will be smaller, so that it can stably fall into a local optimal solution. At the same time, the learning rates required for various machine learning problems are not well set and require repeated debugging. For the optimal learning rate setting, a common way is to test the training loss value of the model at different values of learning rate. However, this method requires a lot of experimentation. The Adam optimization algorithm [38] is an adaptive parameter adjustment algorithm, which is an extension of the SGD algorithm. The Adam algorithm is different from the traditional SGD: SGD maintains a fixed value of the learning rate to update all parameters, and the learning rate does not change during the training process; Adam calculates independent adaptive learning rates for different parameters by calculating the first-order moment estimation and second-order moment estimation of the gradient. Adam algorithm has the advantages of fast calculation speed, suitable for non-stationary target, invariance of gradient diagonal scaling, etc. It is very suitable for solving parameter optimization problems with large-scale data. In the model training process, we use the Adam algorithm to automatically adjust the learning rate.

First, we wanted to explore the best compression performance of the proposed model. For neural networks, increasing the number of training iterations can often improves network performance, so we tested the efficiency of different numbers of training iterations to the performance of the model. The experimental data was the training set and test set of Node 7. During the test, we averaged the summation of PRD value and SNR value of all mini-batches of the test set as the final result. The number of sensing data in a mini-batch was 120. The number of mini-batches of the training set of Node 7 was 289, and the number of mini-batches of the test set of Node 7 was 73. We report the average of PRD value and SNR value of the proposed model on the test set. The results are shown in Figure 5.

At the beginning, increasing the number of training iterations can greatly improve the performance of the model. However, when the number of training iterations reaches 50 and above, the speed of the performance improvement of the model becomes slow. At this time, a large increase in the number of training iterations can only result in a small model performance improvement. In our experiments, when the number of training iterations reached 500, the proposed model achieved the best performance on the test set. The average of reconstruction error reached 0.0678 °C, the average PRD value was 2.3711%, and the average SNR value was 32.51 dB. When the number of training iterations exceeds 500, the performance of the proposed model on the test set begins to deteriorate, because the model over-fits the training data at this time, and the generalization ability of the model decreases, that is, the model has over-fitting. Increasing the number of training iterations will increase the computational consumption of the model. In practical applications, we believe that the optimal number of training iterations is 50. When the number of training iterations is 50, the average PRD value of the model is 3.6915%, and the average of reconstruction error is 0.0973 °C. Although this result is not the optimal result for the proposed model, compared with 500 training iterations, the model performance has not decreased too much, and the training time and computation consumption have been reduced by 10 times. Since the proposed model needs to be calculated on the sensor node, we recommend setting the number of training iterations to 50 in practical applications.

Figure 6 shows the reconstructed data and the original data of Node 7 for the proposed model, with the number of training iterations of 500. We reported 6000 sample points in the test set. The proposed model has high reconstruction accuracy, and the reconstructed data can closely approximate the trend and value of the original data. Although the model is trained to input data in the form of a mini-batch with 120 sample points, after the feature extraction by multiple convolution kernels of different sizes, the proposed model can still fit the approximate values of all the sample points separately, avoiding the reduction of the fitting performance of the model for a single sample point due to the large size of input data. The black line in Figure 6 shows the reconstruction error curve of the Node 7 test set samples. We find that the reconstruction error of the proposed model mainly comes from the parts of the original sample data, the value of which changes drastically. These cases are not well learned by the proposed model because of the low number and probability of occurrences in the original data sample points. We recorded the reconstruction results for all sample points in the Node 7 test set, and the number of sample points is 8520. For all sample points in the Node 7 test set, the maximum value of the proposed model reconstruction error was 1.2301 °C, the minimum value was less than 0.0001 °C, and the average value was 0.0678 °C. For these 8520 sample points, the reconstruction error value of the proposed model for most sample points was less than 0.1 °C, the number of samples with reconstruction error exceeding 1.0 °C was only 18, and the number of samples with reconstruction error exceeding 0.1 °C was only 868.

In this experiment, we used IBRL to compare the performance of our model with other compression algorithms. The results are shown in Table 2. Since the stream data length of the CS algorithm cannot be too long, we selected 40,000 data points of Node 7 for performance testing of the CS algorithm. We divided this data into 8 segments with a segment length of 5000. We averaged the results of all segments as the final result of the CS algorithm and set the CR to 10. DPCM-o algorithm uses the Huffman’s algorithm to generate a dictionary and then compresses the sensing data [16]. LTC algorithm generates a set of line segments which form a piecewise continuous function. This function approximates the original dataset in such a way that no original sample is farther than a fixed error e from the closest line segment [16]. We set e to 130. The experimental methods and results of the Stacked RBM-AE model are from the literature [30]. At the same time, we tested the Stacked RBM-VAE model by using the same network structure and test methods as the Stacked RBM-AE model. The variational part of the Stacked RBM-VAE model is similar to the CBN-VAE model. We referred to Reference [31] to design the CNN-AE model and set the network parameters and training details in the experiment to be the same as the CBN-VAE model. We also tested the compression performance of CBN-VAE at different compression ratios. In Table 2, we denote CBN-VAE, CBN-VAE-b and CBN-VAE-c. Table 3 shows the results of the performance of our model on different datasets. The results show that the proposed model can achieve higher CR value and higher SNR value than other compression algorithms. At the same time, the proposed model can achieve efficient compression performance for different categories of data in different datasets.

### 3.3. Transfer Learning

We believe that generalization performance is an important indicator for compression models. For the neural network model, the generalization performance of the network is also called the transfer learning ability. A common way to test the transfer learning ability of neural network is to test the trained neural networks by using different datasets. If a neural network trained with a single dataset can still achieve good results on other datasets, we believe that the neural network has a strong transfer learning ability. In this experiment, we first used the data of Node 7 to train the model and get the model parameters of the model for Node 7. Then, we used the model parameters of Node 7 to initialize the model and test the compression and reconstruction performance of the model with Node 7 parameters for all nodes. At the same time, we trained each node separately to obtain the model parameters corresponding to each node. For all nodes, we set the number of training iterations of the model to 500, and we report the average PRD value, the average SNR value and the average reconstruction error of the model with different parameters on each node. These average values are for all sample data for each node’s test set. Figure 7 and Figure 8 show the average SNR values and the average reconstruction error of models with different model parameters for all nodes, separately. For Node 5 and Node 45 without sample data, we set the average SNR value and the reconstruction error to 0.

The experimental results show that the proposed model has good compression performance for all nodes. The optimal compression performance of the model is obtained at Node 2, with the average SNR value of 38.98 dB and the average reconstruction error of 0.0387 °C. We can be seen from Figure 7, the average SNR value of the model is not significantly reduced by changing the model parameters. For all nodes, the maximum value of the digital difference in the average SNR value between the blue box and the ‘✴’ symbol in Figure 7 is 4.24 dB, and the minimum value is 0.01 dB. The average SNR values of the model which use the parameters of node self are both more than 30 dB. The high SNR value indicates that the proposed model can better recover useful information from the original data. In Figure 7, we use the upper and lower boundaries of the blue box to represent the confidence intervals of the average SNR values for the corresponding nodes. For most nodes, the average SNR values of the model which using the parameters of Node 7 are all within the confidence interval of the node. This proves that even if the node are not individually trained, the model trained with the Node 7 data can be directly used for data compression of other nodes with a small compression performance reduction. The hidden mathematical features learned by the proposed model are common to the same categories of data in the vicinity. For the reconstruction error of all nodes, most of the model reconstruction errors which using the parameters of node self are below 0.1 °C. For the model that uses the parameters of node self, its compression performance is generally better than using the model parameters of Node 7, but it does not improve too much. For all nodes, the minimum and maximum errors of the digital difference between the data reconstruction of red line and blue line in Figure 8 are 0.0003 °C and 0.067 °C, respectively. For those nodes located near Node 7, such as Nodes 4–10, the reconstruction errors are not significantly reduced by use the model parameters that do not correspond to the node self. This results prove that our algorithm has good transfer learning ability. When the model is applied, we can train only one node’s model and then apply the model to all nodes. This can further reduce the computational consumption of the node model training.

### 3.4. Fault Detection and Anti-Noise Analysis

Since many sensor nodes work outdoors, some fault data and noise data will inevitably be collected during the data collection process. Because there are too many factors affecting the accuracy of data collection in the natural environment, anti-noise capability is essential for a WSN data compression model. In this experiment, we use the proposed model to compress the original data with fault, and observe the data reconstructed by the model. We first use the original sample data of Node 7 to train the model, then use the fault injection method [39] to add different numbers of fault data to the original sample data of Node 7, and use these original samples with fault data to test the compression performance of the trained model. We call this process the anti-noise ability analysis of the model. At the same time, we explored the fault detection ability of the proposed model, that is, whether the model can identify fault data that is different from the original sample data. We equate the fault detection problem with a binary classification problem, the category of the data is judged by the label. Specifically, we record the index of the fault data in the data sample when adding the fault data. When testing, we calculate the fault detection ability of the model based on the fault data index judged by the model.

We inject three data faults into the IBRL: noise fault, short-term fault, and fixed fault. We first calculate the number of fault data we want to inject, and then we randomly select the corresponding number of sample points from the original data and inject data faults into these sample points. The injection of noise fault is achieved by adding a normally distributed random number $\;\varepsilon =N\left(0,\sigma^{2}\right)$ to all the selected sample points. The standard deviation σ of noise  ε  is three times the standard deviation of normal data:(9)$$x_{f}=x_{o}+\varepsilon$$
where $x_{f}\;$ is the fault data, $x_{o}\;$ is the selected sample point of original data.

Short-term fault injection is achieved by increasing the amplitude of the selected sample point by f times (f is a constant). In this paper, f is set to  ±0.25. We set the value of f by setting a random number ρ to ensure that the two calculation probabilities are equal when calculating $\;x_{f}$:(10)$$x_{f}=\{\begin{array}{c} x_{o}+0.25*x_{o},\;\;\rho <0\\ x_{o}-0.25*x_{o},\;\;\rho \geq 0 \end{array}$$

Fixed fault injection is to set the selected sample point to a fixed value:(11)$$x_{f}=x_{o}^{1}$$
where $x_{o}^{1}\;$ is the original data of node1 with the same index point position as the selected sample point.

We use three evaluation indicators commonly used in the binary classification problem to measure the fault detection performance of the model:  precision, recall and $F_{1}\;score$. The precision refers to the ratio of the number of samples that the model prediction is correct to the number of samples that the model predicts to be true.
(12)$$precision=\frac{TP}{TP+FP}$$
where TP (True Positive) refers to the quantity that predicts the true positive as a positive. FP (False Positive) refers to the quantity that predicts the true negative as a positive.

The recall refers to the ratio of the number of samples that the model prediction is correct to the number of true samples.
(13)$$recall=\frac{TP}{TP+FN}$$
where FN (False Negative) refers to the quantity that predicts the true positive as a negative.

For the classification model, separate high precision and separate high recall do not indicate the classification performance of the model. In general, the increase of precision will cause the decrease of recall. Similarly, the increase of recall  will cause the decrease of precision. We want both precision and recall  to have higher values for a classification model. The $F_{1}\;score$ is the harmonic mean of the precision and recall, with a maximum of 1 and a minimum of 0. The larger the $F_{1}\;score$ value of the classification model, the better the classification ability of the model.
(14)$$F_{1}\;score=\frac{2*precision*recall}{precision+recall}$$

We selected 2400 sample points from the test set of Node 7 data, and injected different numbers of three fault data into these sample points respectively. We used these sample points with fault to test the fault detection ability of the model which trained by using the original data of Node 7. We used the reconstruction error to determine which class the sample points are divided into. We recorded the mean $e_{mean}\;$ and standard deviation $e_{std}\;$ of the model’s reconstruction error for original data with no data failure. When using the data with data failure test, if the reconstruction error value of the sample point is at $\left[e_{mean}-3*e_{std},\;e_{mean}+3*e_{std}\right]$, we judge that the sample point is normal data, otherwise, we judge that the sample point is fault data. The experimental results of the fault detection ability of the model are shown in Figure 9. We record the corresponding fault data ratio of the model which has the maximum $F_{1}\;score$ for different data faults. Figure 10 shows the data reconstruction and the anti-noise ability of the model when the model has the maximum $F_{1}\;score$ for different data faults.

When we inject the noise fault into the original data, the $F_{1}\;score$ of the model increases as the ratio of fault data increases. This shows that the model’s ability to detect noise faults is also increasing. For different numbers of noise fault data, the recall of the model is always above 90%, which means that the model can always detect most of the sample points with data fault. However, the low precision value of the model indicates that the model also classifies some normal sample points as fault data. The low precision value and high recall value indicate that the model classifies a lot of data as fault data, which contains most of the real faults and a large amount of normal sample points. As the ratio of fault data increases, the precision value also increases slowly, indicating that the detection accuracy of the model for normal sample points is also improved, that is, the ratio of classifying normal sample points to fault data is reduced. The results analysis of short-term fault are the same as the noise fault, but the ability of the model to detect short-term faults is generally worse than that of noise faults. Figure 10b,c show the reconstruction results of the model for noise fault and short-term fault, respectively, with the ratio of fault data is 50%. We can see that after adding the noise fault with the ratio of fault data is 50%, most of the original information of the sample points is covered by noise. At this time, the reconstruction of the proposed model can only use the hidden features learned from the original data, and cannot use the data with noise fault. At the same time, due to the high amplitude of the noise data, the reconstructed data amplitude of the model is also changing. However, the amplitude change of the noise data is not a fixed value, and the reconstructed data of the model will only move closer to the value where the amplitude changes greatly. Unlike noise faults, short-term faults change the amplitude of sample points by adding a fixed value. Therefore, we see that after adding the short-term fault with the ratio of fault data is 50%, the trend of the original data is maintained, but the amplitude changes drastically. At this time, the model reconstruction is affected by the data amplitude, which causes the reconstructed data value of the model to fluctuate widely around the original data value. The amplitude of the fault data for the noise fault and the short-term fault both exceeds 10 °C, but the amplitude of the fault data in the short-term fault is more severe than the noise fault, so the model has better detection effect on the noise fault than the short-term fault. For the noise fault with the ratio of fault data is 50%, the $F_{1}\;score$ value of the model is 73.40%, and the average reconstruction error is 1.6577 °C. For the short-term fault with the ratio of fault data is 50%, the $F_{1}\;score$ value of the model is 71.99%, and the average reconstruction error is 2.9828 °C. For fixed faults, as the ratio of fault data increases, the detection ability of the model shows a slow downward trend, but it does not decrease too much. For fixed faults with different ratios, the difference between the precision value and the recall value of the model is small, and there is no high recall value and low precision value phenomenon similar to the noise fault and the short-term fault. The optimal $F_{1}\;score$ value of the model is close to the optimal $F_{1}\;score$ value of the noise fault and the short-term fault, which indicates that the model can maintain the detection ability of the $F_{1}\;score$ value exceeding 70% for different faults. Figure 10d shows the reconstruction results of the model for fixed fault with the ratio of fault data is 5%. The fixed fault is the change of original sample to the data of other nodes. The test model is trained by the original data of Node 7, and the model has good transfer learning ability. For other nodes, the hidden mathematical features are similar to those of Node 7, so the model does not recognize the added other node data very well. Therefore, the ratio of fault data is 5% when the model achieves the best detection ability. We can be seen from Figure 10d, as the ratio of fixed fault data increases, the trend of sample data will approach the data of other nodes. Therefore, as the ratio of fault data increases, the detection ability of the model shows a slow downward trend, but because of the transfer learning ability of the model, the detection ability is not much reduced. When the ratio of fixed fault data is small, these fixed faults are noise compared to the original sample, so the model can identify these fixed faults well. For the fixed fault with the ratio of fault data is 5%, the $F_{1}\;score$ value of the model is 73.08%, and the average reconstruction error is 0.0861 °C. The experimental results show that the proposed model has good fault detection ability and anti-noise ability. Even if the fault data is inevitably mixed when collecting data, most of the fault data will be avoided when the model is reconstructed.

### 3.5. Energy Analysis and Optimization

The computational consumption required for model compression directly affects the application of the model and the lifetime of the sensor nodes. We want to minimize the computational consumption of the model while maintaining compression performance. We compare several different data compression neural network models from the number of network parameters, network computation complexity, training time, calculation time, etc. The experimental numerical results are shown in Table 4.

Compared with other neural network models, the CBN-VAE model has the shortest compression time, the minimum number of floating-point operations (FLOPs) and the number of parameters. Using the CBN-VAE model to compress a sensing data sequence with the number is 120 needs 13,917 floating-point calculations (including multiplication and addition). At the same time, we test the single floating-point calculation speed of STM32F407 with Float Point Unit (FPU). When using the FPU, STM32F407 takes 0.072 μs to execute a single floating-point calculation (multiplication and addition). In theory, using the FPU, STM32F407 will take 1.01 ms to complete the floating-point calculation of the model compression. For the sensing node, using STM32F407 as the microcontroller unit (MCU) may result in excessive power consumption of the device, so we also tested the single floating-point calculation speed of the STM32L4 with very low-power. Using the FPU, STM32L4 will take 2.53 ms to complete the floating-point calculation of the model compression.

For the use of the CBN-VAE model in WSN, we propose two solutions. Because the reconstruction of the compressed data must be done by the server, we only discuss the implementation of the compression process. One solution is that the training process of the model is done at the server, and then the server sends the model parameters to the corresponding sensor node. The node uses these parameters to initialize the model and perform data compression. The disadvantage of this solution is that once the parameters of the model are trained, it is difficult to update afterwards because the parameter update requires retraining the model with an amount of new original data. The advantage is that the training process is completed by the server, and the computational consumption of the node can be greatly reduced. Another solution is that the training process of the model is completed by the sensor node itself, and then the node sends the model parameters to the server, and the server uses the parameter to reconstruct the compressed data. The advantage of this solution is that the model parameters can be updated at any time. The disadvantage is that the computational consumption of the training process is provided by the node.

For neural networks, the model parameters are usually redundant. To further streamline the model, we use the neuron pruning method to further reduce the number of parameters of the model and the calculate consumption. Compared to other neuron pruning methods, we believe that pruning neurons must be guided by the importance of this neuron to the entire neural network. We equate the network parameter pruning problem with the neuron classification problem, which divides all neurons in the network into two categories: prunable and non-prunable. Specifically, we apply the classification idea of the decision tree to classify all neurons in the network, then remove neurons with category prunable, and finally restore the network ability through iterative fine-tuning and pruning. This approach is structured pruning without sparse convolution kernels. Algorithm 2 shows the process of pruning neurons. The neuron pruning process is shown in Figure 11.

We evaluate the importance of each neuron in the trained network. For the neuron $N_{i,j}$, the importance score of $N_{i,j}$ is calculated as follows:(15)$$NIS_{i,j}=\frac{ACC_{p}}{ACC_{o}}*100$$
where $\;ACC_{o}$ is the reconstruction accuracy of original network, $ACC_{p}$ is the reconstruction accuracy of original network when the parameter value of $N_{i,j}$ is 0. We constructed a matrix NIS to storage the neuron importance score. We first calculate the importance scores for all neurons, then multiply the prune rate and the number of weight parameters to get the number of parameters we need to prune. When pruning, we set the value of the neuron parameter corresponding to the smaller importance score to 0 according to the number of parameters to be pruned. After pruning, we retrain this simplified model to fine-tune the weights and iteratively the prune and retrain process until the model performance returns to the original performance.

**Algorithm 2:** Pruning Neurons.1: **Input:** The model weights **W**, the prune rate *α*, the number of prune iterations *iter*2: Get m which is the number of elements in **W**3: Get the number of **W** that need pruning n=α∗m4: Calculate the importance of neurons, get the neuron importance score matrix NIS5: Sorting NIS from large to small6: Get the prune threshold of neuron importance score thr=NIS[n]7: **While**
*i* < *m*
**do**8:  **if**
NIS[i] < *thr*
**then**9:  **W**[i] = 010: **end**11: i++12: **end**13: **While**
*k* < *iter*
**do**14:  Retrain the model to update parameters W according to Algorithm 115:  Execute step 2–1216:  k++17: **end**

We compared the pruning results of our neuron pruning method with other common methods at the same pruning rate. We record the reconstruction error of the model at different pruning rates. The results of model reconstruction error are shown in Figure 12. The test model is trained by the data of Node 7. For each pruning method, we retrain the pruned model. The number of iterations of fine-tuning and pruning is 5.

Our pruning method has obvious advantages over other methods. When the pruning rate is 50%, the model reconstruction error of our method is 0.0971 °C, the model reconstruction error of *Random* and *Mag* are 0.6145 °C and 0.3624 °C respectively. When the pruning rate is 80%, the model reconstruction error of our method is only 0.3032 °C, and the reconstruction error of *Random* and *Mag* exceeds 1.5 °C. The results show that our method can accurately identify redundant neurons in the network. In our experiments, using our method to prune 40% of the model parameters does not affect the reconstruction accuracy of the model. Using our method to prune the model can further reduce the parameters and computation consumption of the model.

## 4. Conclusions

In this paper, we propose a neural network compression model with efficient convolutional structure to compress the sensing data of WSNs. Compared with other neural networks, the model has very few parameters and calculations and better compression performance. We detail our efficient convolution structure and methods for reducing parameters and computation consumption when designing the model. We use different real WSN datasets to evaluate the compression performance of the model. At the same time, we compare the model with other algorithms. The results show that our model has higher CR and reconstruction accuracy than other compression algorithms. We also evaluate the model from the aspects of transfer learning, fault detection and anti-noise ability. The evaluation results prove the excellent ability of our model. We perform energy analysis on our model and compare it with other neural network models to demonstrate the effectiveness of our model in reducing energy consumption. Finally, we give a model pruning method to further reduce the parameters and computation consumption of the model.

The results of pruning indicate that there is redundancy in our model. In the experiment, we found that the redundant parameters are mainly concentrated in the fully connected layer. In the next work, we will study how to further reduce the redundancy of the model from the aspects of structure and calculation. A good idea is to replace all fully connected layers with convolutional layers, but we suspect that this may lead to a decline in model performance and an increase in model calculations, and the specific results require us to actually verify. At the same time, we would like to further improve the model by referring to the idea of depth-wise separable convolution in MobileNets [40].

## Figures and Tables

**Figure 1 sensors-19-03445-f001:**
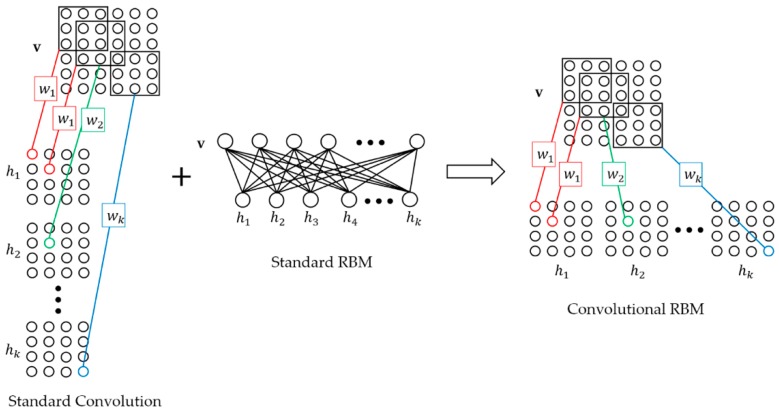
Convolutional Restricted Boltzmann Machine (CRBM).

**Figure 2 sensors-19-03445-f002:**
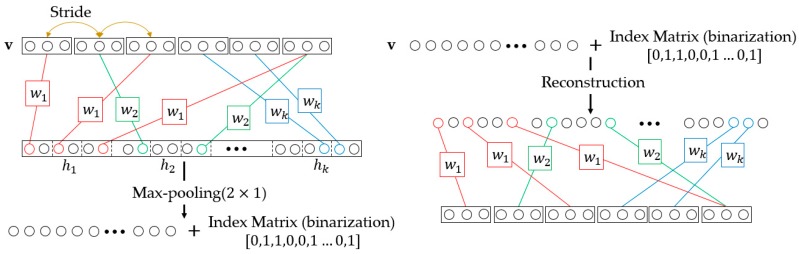
Downsampling-Convolutional RBM.

**Figure 3 sensors-19-03445-f003:**
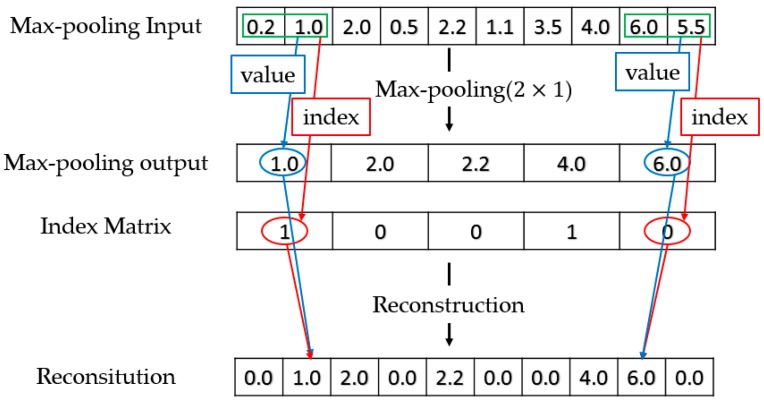
Max-pooling and Reconstruction.

**Figure 4 sensors-19-03445-f004:**
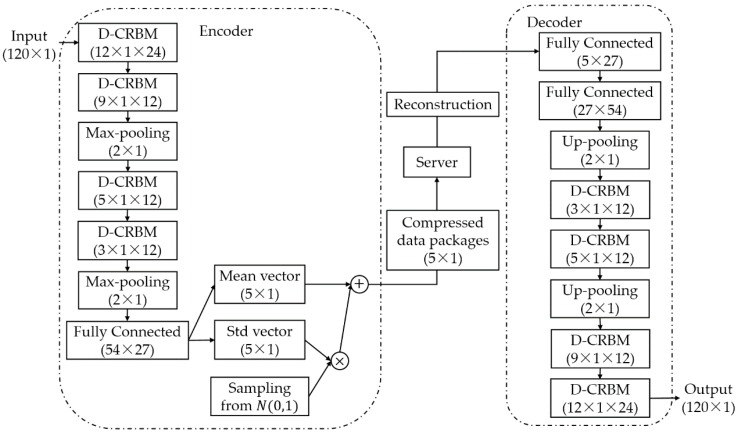
CBN-VAE model. The contents in the parentheses of the network layer (D-CRBM and Fully Connected) are the size of the layer kernel, and the contents in the parentheses of the vector (input and output, etc.) are the size of the vector. For Max-pooling and Up-pooling layers, the contents in the parentheses are the size of the sampling.

**Figure 5 sensors-19-03445-f005:**
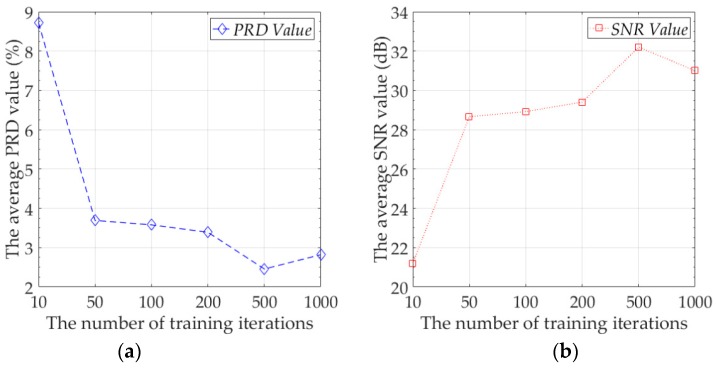
(**a**) The average PRD value under different number of training iterations, (**b**) The average SNR value under different training iterations.

**Figure 6 sensors-19-03445-f006:**
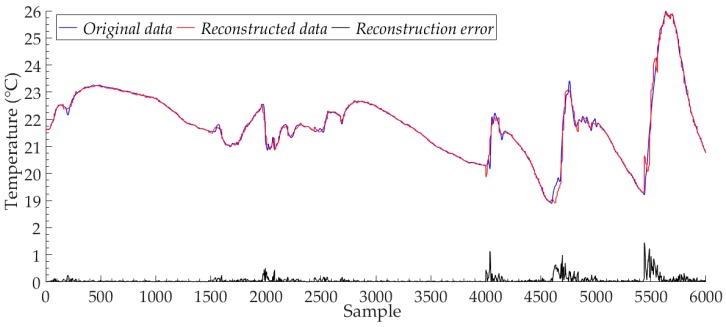
The reconstructed data and the original data of Node 7.

**Figure 7 sensors-19-03445-f007:**
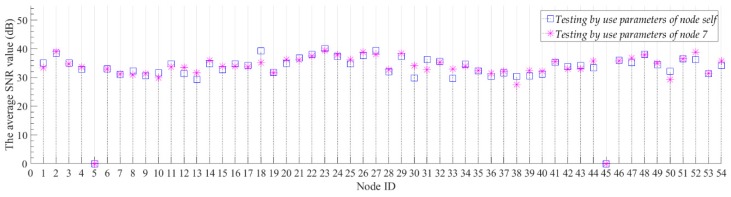
The average SNR values of models with different model parameters for all nodes. The blue box denotes the average SNR value of the model which testing by use the parameters of node self. The upper and lower boundaries of the blue box correspond to the plus or minus 1 dB range of the result value. The ‘✴’ symbol denotes the average SNR value of the model which testing by use the parameters of Node 7.

**Figure 8 sensors-19-03445-f008:**
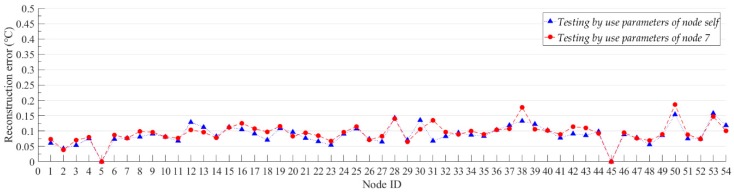
The reconstruction error of models with different model parameters for all nodes. The red line shows the reconstruction error of the model which testing by use the parameters of Node 7. The blue line shows the reconstruction error of the model which testing by use the parameters of node self. The reconstruction error is the average value of all sample points in node.

**Figure 9 sensors-19-03445-f009:**
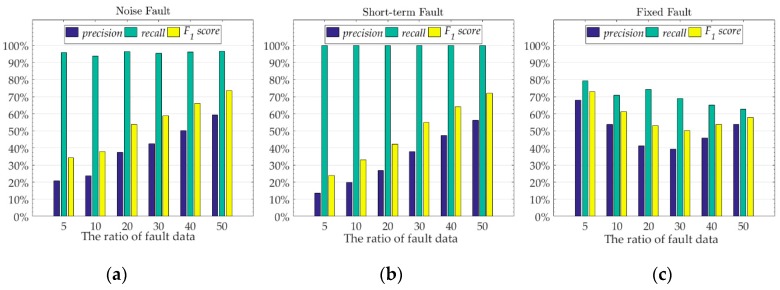
The fault detection performance of the model under different fault data ratios for three different fault. (**a**) noise fault, (**b**) short-term fault, (**c**) fixed fault. The value of the *x*-axis is the ratio of the number of fault data we add to the original sample to the total number of original sample data.

**Figure 10 sensors-19-03445-f010:**
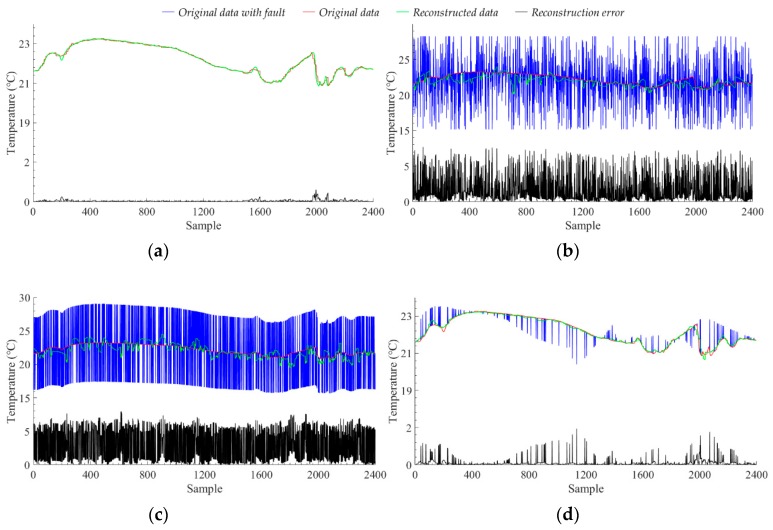
The data reconstruction results of the model for different data faults. (**a**) The original data without fault, (**b**) The original data with noise fault and the ratio of fault data is 50%, (**c**) The original data with short-term fault and the ratio of fault data is 50%, (**d**) The original data with fixed fault and the ratio of fault data is 5%.

**Figure 11 sensors-19-03445-f011:**
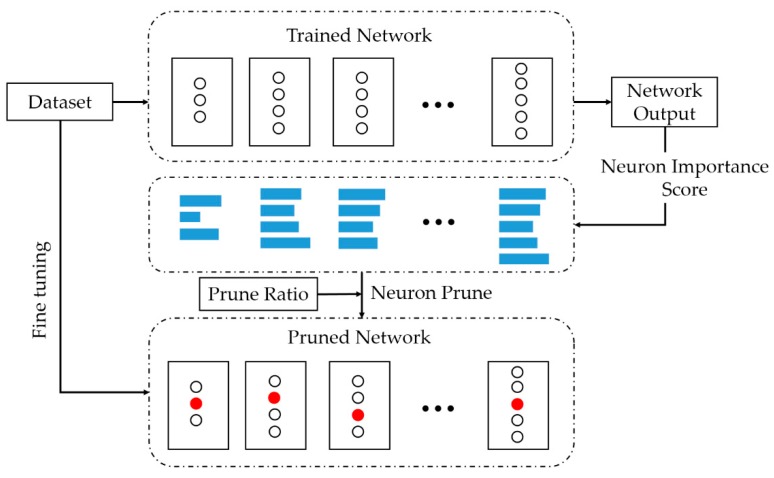
The neuron pruning process. The white ‘〇’ denotes the neuron in the network, the length of the blue rectangle represents the importance score of the corresponding neuron, and the red ‘〇’ represents the neurons that are pruned in the network.

**Figure 12 sensors-19-03445-f012:**
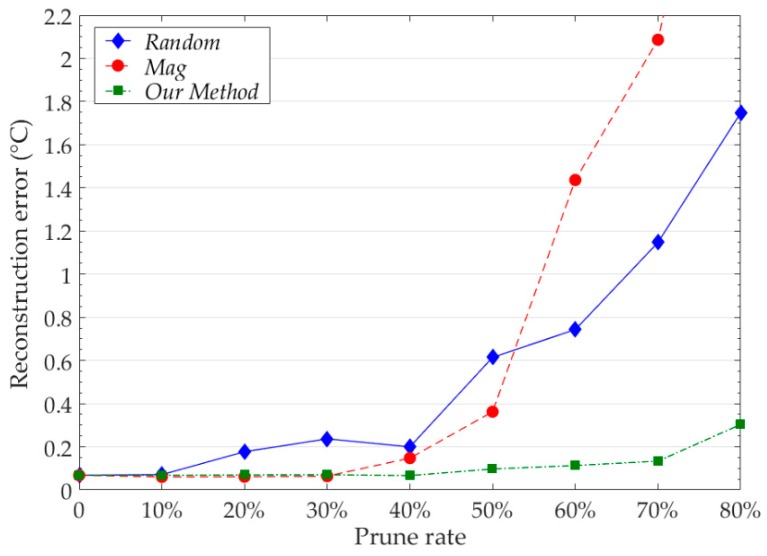
The pruning results of model at different pruning rates for different pruning methods. *Random* denotes the random pruning. *Mag* denotes the magnitude based pruning. The reconstruction error is the average value of all sample points in Node 7.

**Table 1 sensors-19-03445-t001:** Details of layers and parameters used for the CBN-VAE model.

No	Layer Name	Kernel Size	Activation Function	No. of Parameters	Output Size
	*Encoder*				
1	D-CRBM	12 × 1 × 24	ReLU	312	120 × 1 × 1
2	D-CRBM	9 × 1 × 12	ReLU	120	84 × 1 × 1
3	Max-pooling (2 × 1)	−	−	−	42 × 1 × 1
4	D-CRBM	5 × 1 × 12	ReLU	72	54 × 1 × 1
5	D-CRBM	3 × 1 × 12	ReLU	48	108 × 1 × 1
6	Max-pooling (2 × 1)	−	−	−	54 × 1 × 1
7	Fully Connected	54 × 27	ReLU	1485	27 × 1
8	Fully Connected	27 × 5	ReLU	140	5 × 1
9	Fully Connected	27 × 5	ReLU	140	5 × 1
	*Decoder*				
10	Fully Connected	5 × 27	ReLU	140	27 × 1
11	Fully Connected	27 × 54	ReLU	1485	54 × 1 × 1
12	Up-pooling (2 × 1)	−	−	−	108 × 1 × 1
13	D-CRBM	3 × 1 × 12	ReLU	48	54 × 1 × 1
14	D-CRBM	5 × 1 × 12	ReLU	72	42 × 1 × 1
15	Up-pooling (2 × 1)	−	−	−	84 × 1 × 1
16	D-CRBM	9 × 1 × 12	ReLU	120	120 × 1 × 1
17	D-CRBM	12 × 1 × 24	ReLU	312	120 × 1 × 1

**Table 2 sensors-19-03445-t002:** Compression performance comparison.

Algorithm	CR	PRD (%)	SNR (dB)	Reconstruction Error (°C)
CS	10	38.40	8.31	1.4143
LTC	13.93	8.45	21.46	0.2904
DPCM-o	13.85	8.00	21.94	0.2342
Stacked RBM-AE	10	10.04	19.97	0.2815
Stacked RBM-VAE	24	9.49	20.45	0.2926
CNN-AE	24	3.80	28.39	0.1075
CBN-VAE	24	2.37	32.51	0.0678
CBN-VAE-b	10	1.70	35.40	0.0454
CBN-VAE-c	40	2.52	31.98	0.0770

**Table 3 sensors-19-03445-t003:** Model performance on different datasets.

Dataset	CR	PRD (%)	SNR (dB)	Reconstruction Error
Argo (temperature)	24	1.18	38.56	0.0984 (°C)
ZebraNet (location/UTM format)	24	7.23	22.82	114.22
CRAWDAD (speed)	24	2.02	33.89	1.0897 (km/h)
IBRL (voltage)	24	6.61	23.58	0.0199 (V)
IBRL (humidity)	24	4.04	27.85	0.5096 (%RH)

**Table 4 sensors-19-03445-t004:** Neural network model specific details comparison. [FLOPs] denotes the floating-point operations of the model compression and reconstruction. [CP-time] denotes the compression time. [FP-time] denotes the compression and reconstruction time. [TR-time] denotes the training iteration time. These results are obtained when the number of mini-batches is 1, and the time results are the corresponding unit time.

Model	FLOPs	No. of Parameters	CP-Time (ms)	FP-Time (ms)	TR-Time (ms)
Stacked RBM-AE	35,850	19,032	0.2113	0.3873	1.0211
Stacked RBM-VAE	36,150	19,234	0.2817	0.4577	1.6549
CNN-AE	772,008	9245	0.9859	1.5493	2.6761
CBN-VAE	27,564	2415	0.1761	1.162	1.7606

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
