# Peer review of "CBN-VAE: A Data Compression Model with Efficient Convolutional Structure for Wireless Sensor Networks"

_sensors, 2019, doi:10.3390/s19163445_

Round 1
Reviewer 1 Report
This paper presents data compression and reconstruction based on CBN--VAE. Basically, the paper is well-written and the results look good compared to the conventional methods. The reviewer has some further suggestions.
On page 11, in line 8, after mentioning about Fig. 6, it is strange that suddenly Fig. 10 is mentioned. Maybe the error can also be shown in Fig. 6.
On page 11, in the 3rd line from the bottom, it is strange to say that their proposed model can achieve higher CR value and lower SNR value.
In Table 3, what is the CR ratio? 24?
In Table 2 and Table 3, it seems that only one CR ratio is considered for their proposed CBN-VAE. Maybe the authors can choose one database to show the performance versus different CR ratio.
In Table 4, the computation complexity including time and FLOPS is evaluated. However, only four of seven in Table 2 are considered. If all of the methods can be evaluated in Table 4, it will be better.
Author Response
Response to Reviewer 1 Comments
Dear reviewer,
Thank you very much for your recognition and your valuable comments. This time, I modify my manuscript according to your suggestion. At the same time, my classmates helped me correct some of the English writing errors and description errors in the manuscript.
For your suggestions, I explained each point in detail as follows, and revised the corresponding content in the manuscript. I hope that you are satisfied with my processing results.
Point 1: On page 11, in line 8, after mentioning about Fig. 6, it is strange that suddenly Fig. 10 is mentioned. Maybe the error can also be shown in Fig. 6. 

Response 1: We add the error curve in Figure 6, and modified the description of the corresponding position. (on line 383 in the new manuscript)
Point 2: On page 11, in the 3rd line from the bottom, it is strange to say that their proposed model can achieve higher CR value and lower SNR value.
Response 2: We modify this error. (on line 419 in the new manuscript)
Point 3: In Table 3, what is the CR ratio? 24? 

Response 3: Yes, the results in the Table 3 are obtained when the value of CR is 24. We add the CR value to the Table 3. (on line 424 in the new manuscript)
Point 4: In Table 2 and Table 3, it seems that only one CR ratio is considered for their proposed CBN-VAE. Maybe the authors can choose one database to show the performance versus different CR ratio. 

Response 4: We added the experiment results corresponding to the different CRs of CBN-VAE in Table 2. (on line 422 in the new manuscript)
Point 5: In Table 4, the computation complexity including time and FLOPS is evaluated. However, only four of seven in Table 2 are considered. If all of the methods can be evaluated in Table 4, it will be better. 

Response 5: Table 4 evaluates the difference in details between our model and other neural network models. The evaluation items in Table 4 are all for the neural network model, and the evaluation results are obtained when the number of input mini-batches of the neural network is 1. For non-neural network algorithms, it is difficult to have the number of mini-batches. Compression algorithms for non-neural network models are also difficult to evaluate using some of the items in Table 4. And for some algorithms, like CS, the results of the algorithm are also subject to other factors, such as reconstruction methods. So we did not add the compression algorithms of other non-neural network models to Table 4 for evaluation.

Reviewer 2 Report
Section 2.1 together with figure 1 is difficult to understand. For my understanding from the text, the CRBM just changes the traditional convolution to 1 output channel. Such a conversion is not a technical contribution because we can just set the number of output channels to 1 in a traditional convolution. In figure 1, the difference between traditional convolution and convolutional RBM is very vague. It just changes the outputs vertically to horizontally. I didn't see any essential difference between them. Some sentences, like "but the weights between the hidden and visible layers are multiple convolution kernels" and "The CRBM replaces the input layer and weights multiplication step with the input layer and multiple convolution kernel convolution calculation", are not understandable. What do you mean by "multiple convolution kernels"? In a traditional convolutional layer, typically we also have multiple convolution kernels.
The algorithm 1 is almost useless because it is just a standard well-known training process. It seems to me that only the loss calculation needs to be highlighted.
In figure 5, it shows that using 1000 iterations produces worse result than using 500 iterations. It is claimed that using 50 iterations is a good choice. Is this a general conclusion or just a specific conclusion for this dataset? If I use a different dataset, does this conclusion still hold?
Some figures are too big and out of the text range.
For table 4, why does the proposed method have higher TR and FP time than the first 2 methods?
Author Response
Response to Reviewer 2 Comments
Dear reviewer,
Thank you very much for your recognition and your valuable comments. This time, I modify my manuscript according to your suggestion. At the same time, my classmates helped me correct some of the English writing errors and description errors in the manuscript.
For your suggestions, I explained each point in detail as follows, and revised the corresponding content in the manuscript. I hope that you are satisfied with my processing results.
Point 1: Section 2.1 together with figure 1 is difficult to understand. For my understanding from the text, the CRBM just changes the traditional convolution to 1 output channel. Such a conversion is not a technical contribution because we can just set the number of output channels to 1 in a traditional convolution. In figure 1, the difference between traditional convolution and convolutional RBM is very vague. It just changes the outputs vertically to horizontally. I didn't see any essential difference between them. Some sentences, like "but the weights between the hidden and visible layers are multiple convolution kernels" and "The CRBM replaces the input layer and weights multiplication step with the input layer and multiple convolution kernel convolution calculation", are not understandable. What do you mean by "multiple convolution kernels"? In a traditional convolutional layer, typically we also have multiple convolution kernels. 

Response 1: We are very sorry that our description is not clear to you. We have modified the description of CRBM. (on lines 165-189 in the new manuscript)
The CRBM is designed by the standard RBM structure, and the example in Figure 1 describes that its calculation of the state of the neuron is different from the standard RBM. Compared to the standard convolution, only the convolution calculation is the same. The CRBM has an undirected graph feature such that its information can be transmitted bidirectionally in the CRBM, which is not possible with standard convolution. The CRBM splicing output channel is designed to meet the undirected graph characteristics of RBM, and the reason is also explained in the new manuscript.
Point 2: The algorithm 1 is almost useless because it is just a standard well-known training process. It seems to me that only the loss calculation needs to be highlighted.
Response 2: We add the details of the loss function calculation, and give the reason for using this loss function. (on lines 279-298 in the new manuscript)
Point 3: In figure 5, it shows that using 1000 iterations produces worse result than using 500 iterations. It is claimed that using 50 iterations is a good choice. Is this a general conclusion or just a specific conclusion for this dataset? If I use a different dataset, does this conclusion still hold? 

Response 3: This conclusion is obtained for the corresponding data set of our experiment. We recommend using 50 iterations because we consider that the model needs to be trained on the nodes, even if the model accuracy is not optimal at this time. We also tested different data sets. For the data set we tested, the model can get good accuracy when the number of training iterations reaches 50, but if you want to get better model accuracy, we recommend increasing the number of training iterations until the model accuracy begins to decline.
Point 4: Some figures are too big and out of the text range. 

Response 4: This is a problem with our manuscript layout, we modify the sizes of figures in the new manuscript
Point 5: For table 4, why does the proposed method have higher TR and FP time than the first 2 methods?
Response 5: For the first two methods, the computational complexity of the reconstructed part is lower, so its TR and FP times will be lower. Because their reconstruction part is just a simple matrix multiplication. For our method, we need to perform deconvolution calculations during reconstruction. This calculation is time consuming in the test framework we use, so our method's TR and FP times will be longer. Because the reconstruction part is done by the server, we only need to ensure that the compression time is as short as possible. The longer TR and FP time is one of the issues that we need to improve later.

Reviewer 3 Report
Paper shows a data compression model for WSN, based on CNN.
In general, paper is well written, with reasonable results. However, the nature of sensors (low energy embedded systems) makes unfeasible the adoption of such technique. To clarify myself, some questions should be answered:
1) Why adopt this method over classical compression techniques? How it compares to plain Huffman (or similar, like correction codes)?
2) What is exactly the power consumption of training/run the network on the node? How it compares to other similar approaches?
3) In which degree the technique impacts the data overhead due to weigths (or topology, or any parameters) tranfers between the server and the nodes?
4) How would you implement this on a real WSN? By the paper, it seems targeted to application level, but data compression and aggregation present some advantages when cross layered (e.g. routing level).
5) One of the good results is the neuron (model) pruning, however there is no comparison with other ANN (and CNN) compression/pruning methods. How your method compares with (for example) plain SqueezeNet usage?
6) In which scenario the method would apply? Usually nodes have a limited well known number of sensors, and a well known range of measurement. In this classical scenario, would your method present any advantage over classical methods?
The conclusion is good, as well the results, however the references are poor. In a quick search, I could find a couple of works about WSN compression, including adaptative and ANN usage.
Bottom line: the work is good, but lacks some comparison with current work.
Author Response
Response to Reviewer 3 Comments
Dear reviewer,
Thank you very much for your recognition and your valuable comments. This time, I modify my manuscript according to your suggestion. At the same time, my classmates helped me correct some of the English writing errors and description errors in the manuscript.
For your suggestions, I explained each point in detail as follows, and revised the corresponding content in the manuscript. At the same time, we enriched our references and results. I hope that you are satisfied with my processing results.
Point 1: Why adopt this method over classical compression techniques? How it compares to plain Huffman (or similar, like correction codes)?
Response 1: We add a summary of coding techniques in the introduction chapter of the new manuscript. (on lines 61-71 in the new manuscript) The biggest difference between this method and the traditional method is the idea of using neural network. At the same time, we consider the compression problem from the mathematical distribution characteristics of the data. We use neural network to learn the distribution characteristics of the data, and reconstruct the data by using the learned distribution. Compared to the encoding method, our method can achieve a higher compression ratio. At the same time our experiments prove that the error of our method is low enough.
Point 2: What is exactly the power consumption of training/run the network on the node? How it compares to other similar approaches?
Response 2: We did not deploy our method to the sensor node for power testing. We calculated the computational complexity of our method and tested the computational performance of different microcontrollers. The actual power consumption is affected by many factors, we can not give a certain value, we can only give a data description from a theoretical point of view. (on lines 610-620 in the new manuscript)
Table 4 shows the comparison of our method with other similar methods. We compare the computational complexity, algorithm parameters, compression time and other aspects to prove the advantages of our algorithm. (on lines 604-608 in the new manuscript)
Point 3: In which degree the technique impacts the data overhead due to weigths (or topology, or any parameters) tranfers between the server and the nodes? 

Response 3: We give the application of our method in the manuscript, we propose two solutions. (on lines 621-633 in the new manuscript) Servers and nodes only need to transfer weights once. At the same time, the weight of our method is also very small.
Point 4: How would you implement this on a real WSN? By the paper, it seems targeted to application level, but data compression and aggregation present some advantages when cross layered (e.g. routing level). 

Response 4: We give the application of our method in the manuscript, we propose two solutions. (on lines 621-633 in the new manuscript)
As you said, our method is at the application level and we don't consider the routing level. But the transfer learning ability of our method also allows our method to have similar utility for data aggregation. The routing level is also the next issue that we need to consider. Thank you for your suggestion.
Point 5: One of the good results is the neuron (model) pruning, however there is no comparison with other ANN (and CNN) compression/pruning methods. How your method compares with (for example) plain SqueezeNet usage?
Response 5: We compared the pruning results of our neuron pruning method with other common methods at the same pruning rate, the comparison results are shown in Figure 12. (on lines 660-676 in the new manuscript)
Sorry, we have not used SqueezeNet and have no experience with SqueezeNet for data compression, so we have not compared our method with SqueezeNet.
Point 6: In which scenario the method would apply? Usually nodes have a limited well known number of sensors, and a well known range of measurement. In this classical scenario, would your method present any advantage over classical methods?
Response 6: Our method are application level and are suitable for most sensing situations. But our method needs as much data as possible to train, so our method hopes to the train sensing data as accurately as possible. The performance of our method is not limited by the number of sensors and the range of measurement. In the same environment, we believe that our approach can achieve higher compression performance than traditional methods.

Reviewer 4 Report
This is an interesting technical paper. The authors’ aims are clear and the approach outlined will be of interest to many involved in the design of models to reduce the communication energy consumption in Wireless Sensor Networks (WSNs)
The paper cites appropriate literature and makes its case well. The abstract identifies the key ideas of the author's work, the scope and results but is too long.
There is a need to insert a section of learned lessons or future research directions. The authors indicate in a single line the next step. It could be of interest to the reader.
The authors should reference the paragraph that is between lines 381-383. It has been literally copied.
They should also review and correctly format references 23 and 25. See section references.
Review the references on line 302: ZebraNet [32] and CRAWDAD [32].
Author Response
Response to Reviewer 4 Comments
Dear reviewer,
Thank you very much for your recognition and your valuable comments. This time, I modify my manuscript according to your suggestion. At the same time, my classmates helped me correct some of the English writing errors and description errors in the manuscript.
For your suggestions, I explained each point in detail as follows, and revised the corresponding content in the manuscript. At the same time, we enriched our references and results. I hope that you are satisfied with my processing results.
Point 1: There is a need to insert a section of learned lessons or future research directions. The authors indicate in a single line the next step. It could be of interest to the reader.
Response 1: We enrich the future research direction of the manuscript. (on lines 690-696 in the new manuscript)
Point 2: The authors should reference the paragraph that is between lines 381-383. It has been literally copied.
Response 2: We add reference marks for the corresponding positions. (on lines 406-411 in the new manuscript)
Point 3: They should also review and correctly format references 23 and 25. See section references.
Response 3: We modify the format of Reference 23 in the original manuscript. (on line 763 in the new manuscript) We verified the format of Reference 25, and we determined that it had no problems, even though it looked strange.
Point 4: Review the references on line 302: ZebraNet [32] and CRAWDAD [32]. 

Response 4: These two datasets are downloaded from a website, and we use the homepage of the website as a reference, so their reference marks are the same.

Round 2
Reviewer 1 Report
The authors have addressed my comments adequately. The reviewer has no more suggestions.
Reviewer 2 Report
I am satisfied with the answers and revisions.